# Growth Charts for Height, Weight, and BMI (6–18 y) for the Tuscany Youth Sports Population

**DOI:** 10.3390/ijerph16244975

**Published:** 2019-12-06

**Authors:** Gabriele Mascherini, Giorgio Galanti, Luciano Massetti, Piergiuseppe Calà, Pietro Amedeo Modesti

**Affiliations:** 1Department of Experimental and Clinical Medicine, University of Florence, 50134 Florence, Italy; 2Institute of Biometeorology, National Research Council, 50145 Florence, Italy; 3Sector "Health and Safety in the Workplace and Special Processes in the Field of Prevention", Directorate of Citizenship Rights and Social Cohesion, Tuscany Region, 50139 Florence, Italy

**Keywords:** overweight, obese, young athletes, percentile, children, adolescent

## Abstract

Overweight during youth is a large-scale public health issue. Engaging in regular physical activity generally reduces weight status. The hypothesis of the study is that organized sport plays an active role in maintaining a correct body weight during youth. The purpose of this study is to trace growth charts by height, weight, and body mass index (BMI) to be applied to the youth sports population. A retrospective study was conducted on 14,700 young athletes (10,469 males and 4231 females) aged between 6 and 18 years from surveillance carried out during the pre-participation screening of sports eligibility. The calculation of the prevalence of overweight and obesity was also performed. The new percentiles for the youth sports population show BMI values at 18 years 21.9 kg/m^2^ for males and 20.7 kg/m^2^ for females at the 50th percentile. The male sample shows 12.3% of the subjects were overweight and 1.5% were obese, while the female sample shows 9.8% are overweight and 1.1% obese. The higher prevalence of excess weight is evident up to 12 years old in both sexes and then gradually decreases. The development of the new specific growth charts for the youth sports population could reduce the risk of error in identifying the correct weight status of young athletes.

## 1. Introduction

A growing prevalence of overweight and obesity has been described in the world population. Part of this increase is attributed to the weight gain experienced in the youth population, which is occurring mainly in developed countries [1].

Worldwide the prevalence of overweight and obesity among children and adolescents aged 5–19 has risen from 4% in 1975 to over 18% in 2016 [2]. In particular, in developed countries 16.9% of males and 16.2% of females were overweight or obese in 1980, compared with 23.8% of males and 22.6% of females in 2013 [3].

Overweight and obesity are major risk factors in adults for a number of chronic diseases including diabetes, cardiovascular diseases, and cancer [4]. Childhood obesity is associated with premature death and disability in adulthood, in addition to increased future risks, obese children experience breathing difficulties, increased risk of hypertension, cardiovascular disease, insulin resistance, and psychological disorders [5].

Physical activity intervention is a useful strategy that can reduce the risk of overweight/obesity in children and adolescents up to 18 years [6,7], and participation in an organized sport is a recognized method to obtain a Body Mass Index (BMI) reduction [8,9]. Participation in youth sports has been positively associated with higher levels of physical activity, [10] however to date, the relationship between youth sport participation and overweight/obesity status is not clear [10,11].

In Italy, since 1982, in order to reduce the risk of sudden death, every person who engages in competitive sports has to undergo a medical examination to obtain eligibility. During pre-participation screening, the athletic population performs a cardiovascular and clinical assessment that includes the evaluation of weight status [12]. The values of height, weight, and BMI recorded during the examination in young athletes are compared with the growth charts developed for the general Italian youth population [13].

The purpose of the study is to verify a lower prevalence of overweight or obesity condition in young people who engage in organized sports compared to the general youth population. Whether this prevalence differs, the linked purpose of the study is to develop for the first time growth charts for height, weight, and BMI in an Italian youth sports population.

## 2. Materials and Methods

A retrospective study was conducted on the data coming from surveillance carried out during the pre-participation screening of sports eligibility. These data came from the regional reference center for Sports Medicine of the Tuscany Region, Italy in the period of 1998 to 2019 inclusive.

### 2.1. Study Population

We have analyzed the data from 14,700 young athletes (10,469 males and 4231 females) aged between 6 and 18 years (mean age 13.9 ± 2.4 for males and 13.1 ± 2.7 years for females). Inclusion criteria for the subjects were to be Caucasian, practice sports at a competitive level, and not to have any contraindications to sports eligibility. Exclusion criteria in the analyses were: having already carried out the same visit and being already included in the study sample at a lower age; an age outside the range of ±6 months compared to the average age of its own stratum; and abnormal values for weight and/or height (5 kg below 3rd or 30 kg above the 97th percentile or 5 cm below 3rd or 5 cm above the 97th percentile).

Sport eligibility certification is 1-year length and generally, it does not coincide with the beginning of the competitive season. Therefore, the evaluations were performed during the regular season. The study was carried out in conformity with the ethical standards laid down in the 1975 declaration of Helsinki. This study is part of a project of the Tuscany Region called “Sports Medicine to support regional surveillance systems”. It was approved by the Regional Prevention Plan 2014–2018 with the number O-Range18. All data has been processed anonymously.

### 2.2. Clinical Evaluation

All children underwent pre-participation screening in accordance with the Italian protocol [14], which includes family and personal history, physical examination, and finally cardiological evaluation. The physical examination focused on the measurement of height and weight by trained personnel, using appropriate equipment (seca gmbh &Co., Hamburg, Deutschland) that has been improved over the years providing ever-greater accuracy. In brief, the subject stood straight, with feet placed together and flat on the ground, heels, buttocks and scapulae against the vertical backboard, arms loose and relaxed with the palms facing medially. The head was positioned in the Frankfurt plane. Body weight was measured in minimal underclothes to the nearest 100 g. BMI was calculated using the formula weight/height ^2^ (kg/m^2^).

### 2.3. Statistical Analysis

#### 2.3.1. Study of the Prevalence of Overweight and Obesity

In order to define normal weight, overweight, or obese status, a subdivision according to BMI, age, and sex were adopted following the classification based on the International Obesity Task Force (IOTF) for children and adolescent [15]. This classification proposes BMI intervals every 6 months: the evaluation of the prevalence of excess weight of the sports sample of the present study was carried out through this subdivision. For a better interpretation, the results are described annually.

#### 2.3.2. Tracing Growth Charts

Smoothed age and gender-specific percentiles (3rd, 50th, and 97th) for height, weight, and BMI were constructed by means of a comprehensive method of smoothing for growth curves using lambda-mu-sigma parameters according to Cole (LMS method) [16,17].

The LMS method summarizes the growth reference curve with three curves representing the median (M), the coefficient of variation (S), and the power to remove skewness from the data (L) by age, and was implemented in the Generalized Additive Model for Location, Scale, and Shape (GAMLSS) package included in R 3.4.1 software for Windows. In the LMS method, GAMLSS parameters and the parameters of Box-Cox power exponential distribution were used for model fitting to data. These reference curves were fitted to the original data and the best fit was used to construct smoothed percentile curves. After the application of the BoxCox power transformation, the data at each age were normally distributed and the points on each percentile curve were defined in terms of the formula: M = (1 + LSz) 1/L where L, M, and S are values of the fitted curves at each age, and z indicates the z score for the required percentile.

## 3. Results

### 3.1. Prevalence of Overweight and Obesity

The age-standardized mean BMI of this young athlete sample is 19.1 kg/m^²^ for females and 19.9 kg/m^²^ for males, resulting in a prevalence of 11.5% overweight, 1.4% of obesity of whole sample. Table 1 and Table 2 show the prevalence of overweight and obesity in young male and female athletes for each year of growth. Between the sexes, there are slight differences in the prevalence of overweight (males 12.3%, females 9.8%) and obesity (males 1.5%, females 1.1%). A similar pattern was verified in both sexes, where the highest prevalence of overweight and obesity is at 8 years, after which these values decrease regularly up to 17 years.

### 3.2. Growth Charts

Table 3 reports the numerical values of percentiles by sex and age for height, weight, and BMI expressed as 3rd, 50th, and 97th percentiles of young athletes. At 18 years of age, the 50th percentile for BMI for males is 21.9 kg/m^2^ while for females it is 20.7 kg/m^2^.

Figure 1 describes graphically the 3rd, 50th, and 97th percentiles of height, weight, and BMI distribution for young athletes.

## 4. Discussion

The first aim of the study was to estimate the overweight and obesity prevalence in a population of young athletes. A recent article published in Lancet in 2017 [1], show that the global age-standardized mean BMI of children and adolescents aged 5–19 years in 2016 was 18.6 kg/m^²^ for females and 18.5 kg/m^²^ for males. Our results derived from a young athlete’s population are in line with these parameters, especially if we consider that they derive from a developed country and those reported at a global level also include developing countries.

Also in 2016, the Italian youth population (aged 6–17 years) had 20.5% for females and 28.6% for males in overweight and obesity status [18]. Our young athletes reported a lower prevalence of overweight and obesity, in particular 10.9% for females and 13.9% for males.

The surveillance system called “Okkio alla salute” detects the weight status of Italian children aged between 8 and 9 years every 2 years. The latest available data refer to 2016 where 26.9% were overweight and obese in the Tuscany Region [19]. In the same age group, the combined prevalence of both sexes in this sample of young athletes is 18%.

The Italian Health Behavior in School-aged Children [20] examined in 2018 in a large sample of 11–15 year old students, that showed males were more likely to be overweight or obese than females (25.1% vs. 15.4% at 11 years-old, 25.0% vs. 14.6% at 13 years, and 25.1 vs. 13.0% at 15 years). Our study also shows a higher prevalence in males than females in the same age groups (16.7% vs. 10.8% at 11 years-old, 14.5% vs. 9.6% at 13 years, and 12.2 vs. 9.7% at 15 years). However, in both sexes, the sample of athletes have a lower excess weight status compared to the sample of students. The results obtained by our sample of young athletes differ from the data already present in the same territory.

However, there is an aspect that joins all the results. There is a tendency to reduce overweight throughout the entire growth process. In fact, the highest prevalence was between 6 to 10 years. From 11 years, this reduction is greater in female than in male peers.

However, the protective role of organized sport with regard to excess weight is not yet clear. A study carried out in Australia [21] showed that there were no associations between sports participation and weight status/adiposity in young people aged between 12 and 17 years. Still following the IOTF guidelines [15], the prevalence of overweight was higher than our results (23%), probably because the sample enrolled did not include only young athletes. A review performed by Lee [10] reports an inconclusive relationship between youth sport participation and obesity status. Despite some studies reporting a BMI reduction with intervention based on a sports program in sedentary and overweight children [9], recent systematic reviews show that once childhood obesity is established it is difficult to achieve a long-term reduction in BMI through multidisciplinary interventions [22].

However, the hypothesis of the present study was to verify a possible protective function, a primary prevention role, that organized sport has on the probability of incurring in an excess body weight during youth. The results obtained in the young athlete sample should confirm this hypothesis.

The prevalence of overweight in the sample of young athletes is still lowering from adolescent age. The 2016 World Health Organization Commission reports that adolescence can be a critical time for excess weight gain [2]. Generally, at this age there is a decrease in sports practice and the beginnings of a greater self-determination in everyday life choices with independent activities outside the home compared to younger children. Also in Italy, the highest level of participation in organized sports activities is at an age between 6 and 10 years with 59.7% of the population. Already in the following age group, those 11–14 years, the females begin to abandon sports, a phenomenon that then consolidates in the age group of 15–17 years [18]. Promoting participation in an organized sport during childhood and being able to continue it during adolescence could predispose to a more active lifestyle even during adulthood [7]. Therefore, being able to continue sports during the transition between childhood and adolescence could reduce the incidence not only of overweight during youth but also during adulthood. Sports educators and professionals should find ways to engage children in sporting contexts and encourage sport participation especially during the adolescent phase.

The second purpose was to propose growth charts for height, weight, and BMI for the first time in a population of Italian young athletes once the prevalence of overweight differed from the general youth population. The growth charts were then processed and these differ from those used in the general population in the same territory [13]. These differences are identifiable especially in 97th percentile for weight, height, and BMI in both sexes. On average, young athletes weigh 2 to 5 kg less and are 3 to 6 cm taller in both sexes. In particular, these differences are greater in the 11–14 age group. BMI has some inherent limitations in accurately establishing the overweight condition with the concomitant excess fat [23]. Children and adolescents may participate in sports that favor a particular body shape, and young athletes that engage in these sports may desire to gain weight and muscle mass. In line with this notion is that some males and females in youth sport could have large lean muscle mass, which could elevate their BMI. On the other side, other sports emphasize a slim or lean physique for aesthetic or performance reasons [24].

The study has the limitation of being retrospective. This influences causality because the weight status recorded is established in advance of the day of the medical evaluation. However, the authors chose this study design to achieve this sample size with such homogeneity among subjects. 

The present study has some strengths. The first is that the subjects were different from each other, they all belonged to the same center, and the evaluation methodology was standardized. The second is the homogeneity of the sample, all the young people practiced organized sports at competitive level. Therefore, the references for height, weight, and BMI presented in this paper are the first to be applied to the population of young athletes aged from 6 to 18 years.

## 5. Conclusions

Young athletes’ references intend to provide pediatricians and sports medicine physician a more specific tool, which refers more precisely to the sporting population and therefore more suitable to adequately monitor the growth of their patients.

The development of these new specific growth charts for the youth sports population could reduce the error risk in young athlete’s weight status identification.

## Figures and Tables

**Figure 1 ijerph-16-04975-f001:**
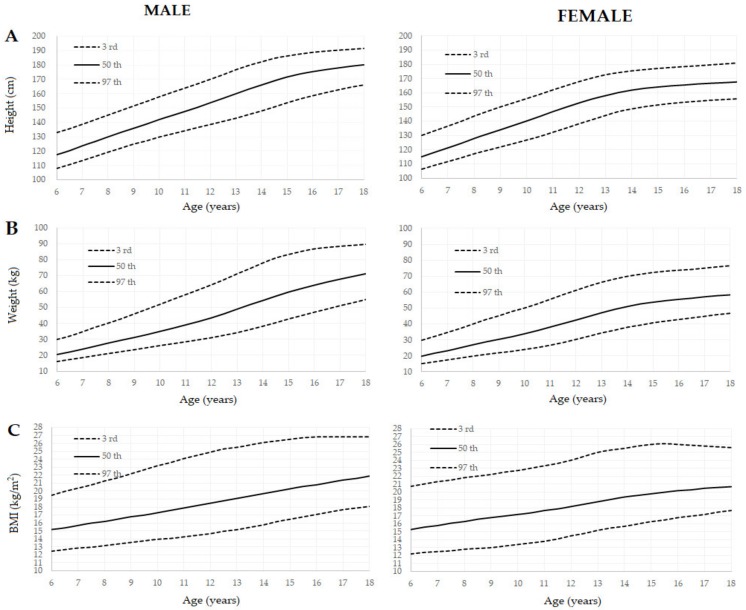
3rd, 50th, and 97th percentiles of height (top-**A**), weight (middle-**B**), and Body Mass Index (BMI) (bottom-**C**) distribution for Italian male athletes (left) and female athletes (right).

**Table 1 ijerph-16-04975-t001:** Prevalence of overweight and obesity in young male athletes divided by age.

Total Sample		Normal Weight		Overweight		Obese	
**Age**	**n.**	**%**	**n.**	**%**	**n.**	**%**	**n.**
6	189	88.9	168	8.5	16	2.6	5
7	135	83.7	113	13.3	18	2.9	4
8	166	80.7	134	16.7	28	2.4	4
9	242	82.6	200	14.9	36	2.5	6
10	436	84.6	369	14.4	63	0.9	4
11	675	83.3	562	15.1	102	1.6	11
12	1567	83.8	1313	14.5	227	1.7	27
13	1631	85.6	1396	12.8	208	1.7	27
14	1618	86.5	1399	12.1	195	1.5	24
15	1611	87.8	1415	10.8	174	1.4	22
16	1524	88.2	1344	10.7	163	1.1	17
17	675	91.4	617	7.7	52	0.9	6
	10,469	86.3	9038	12.3	1274	1.5	157

**Table 2 ijerph-16-04975-t002:** Prevalence of overweight and obesity in young female athletes divided by age.

Total Sample		Normal Weight		Overweight		Obese	
**Age**	**n.**	**%**	**n.**	**%**	**n.**	**%**	**n.**
6	122	82.8	101	10.7	13	6.6	8
7	88	77.3	68	14.8	13	6.8	6
8	141	75.9	107	16.3	23	3.5	5
9	192	83.3	160	14.6	28	2.1	4
10	295	86.1	254	13.6	40	0.3	1
11	426	89.2	380	10.6	45	0.2	1
12	613	88.2	541	11.1	68	0.7	4
13	626	90.4	566	9.1	57	0.5	3
14	590	91.3	539	8.0	47	0.7	4
15	495	90.3	447	8.3	41	1.4	7
16	436	95.1	415	6.0	25	0.5	2
17	207	93.7	194	5.8	12	0.5	1
Total	4231	89.1	3772	9.8	413	1.1	46

**Table 3 ijerph-16-04975-t003:** Height, weight, and Body Mass Index (BMI) growth norms, expressed as 3rd, 50th, and 97th percentiles of young athletes.

	Male									Female								
	Height (cm)			Weight (kg)			BMI (kg/m^2^)			Height (cm)			Weight (kg)			BMI (kg/m^2^)		
Age (yrs)	3rd	50th	97th	3rd	50th	97th	3rd	50th	97th	3rd	50th	97th	3rd	50th	97th	3rd	50th	97th
6	107.8	117.5	132.8	15.9	20.5	30.1	12.5	15.2	19.5	106.1	115	130.3	15.0	19.8	29.9	12.2	15.3	20.7
6.5	110.5	120.3	135.5	17.2	22.1	32.2	12.7	15.4	20.0	108.9	118.2	133.6	16.2	21.7	32.3	12.4	15.6	21.0
7	113.3	123.7	138.6	18.4	23.9	34.9	12.9	15.7	20.4	111.5	121.3	136.8	17.4	23.2	34.8	12.5	15.8	21.3
7.5	116.2	126.7	141.8	19.7	25.8	37.8	13.0	16.0	20.8	114.1	124.4	140.1	18.6	25.1	37.3	12.6	16.1	21.5
8	119.2	129.9	145.0	21.0	27.7	40.3	13.2	16.2	21.3	117.0	127.9	143.8	19.8	27.0	40.1	12.8	16.3	21.8
8.5	122.0	133.0	148.2	22.2	29.5	43.0	13.4	16.5	21.7	119.4	131.0	147.1	20.9	28.8	42.9	12.9	16.6	22.0
9	124.9	135.9	151.4	23.4	31.2	46.0	13.6	16.8	22.2	121.8	134.0	150.2	21.9	30.4	45.2	13.0	16.8	22.2
9.5	127.1	138.8	154.5	24.7	33.0	49.0	13.8	17.0	22.7	124.2	137.1	153.1	22.8	32.0	47.9	13.2	17.0	22.5
10	129.8	142.0	157.8	26.0	35.0	52.0	14.0	17.3	23.2	126.7	140.2	156.0	24.0	33.9	50.1	13.4	17.2	22.7
10.5	132.0	144.8	160.9	27.2	37.0	55.1	14.1	17.6	23.6	129.2	143.4	159.0	25.2	35.9	52.7	13.6	17.4	23.0
11	134.2	147.6	163.9	28.4	39.1	58.1	14.3	17.9	24.1	132.2	146.8	162.1	26.7	38.1	55.6	13.8	17.7	23.3
11.5	136.5	150.5	166.9	29.7	41.2	61.1	14.5	18.2	24.5	135.1	149.9	165.0	28.4	40.3	58.5	14.1	17.9	23.6
12	138.6	153.7	170.1	31.0	43.5	64.2	14.7	18.5	24.9	138.2	152.9	167.9	30.4	42.5	61.2	14.5	18.2	24.0
12.5	140.8	156.8	173.3	32.6	46.1	67.5	15.0	18.8	25.3	141.1	155.7	170.4	32.4	44.8	63.9	14.8	18.5	24.5
13	143.0	160.0	176.8	34.2	49.0	71.1	15.2	19.1	25.5	144.0	158.0	172.6	34.5	47.1	66.3	15.2	18.8	25.0
13.5	145.6	163.2	179.8	36.2	51.8	74.4	15.5	19.4	25.8	146.8	160.2	174.1	36.2	49.2	68.3	15.5	19.1	25.3
14	148.1	166.1	182.3	38.3	54.4	77.9	15.8	19.7	26.1	148.6	161.9	175.3	38.1	51.0	70.0	15.7	19.4	25.5
14.5	150.9	169.0	184.8	40.5	57.1	81.0	16.2	20.0	26.3	150.2	163.2	176.2	39.4	52.6	71.2	16.0	19.6	25.8
15	153.8	171.7	186.4	42.9	59.7	83.2	16.5	20.3	26.5	151.4	164.1	177.0	40.9	53.7	72.4	16.3	19.8	26.0
15.5	156.5	173.8	187.8	45.0	61.9	85.1	16.8	20.6	26.7	152.5	164.9	177.7	42.0	54.7	73.2	16.5	20.0	26.1
16	158.8	175.4	189.0	47.2	64.0	86.7	17.1	20.8	26.8	153.3	165.5	178.3	43.0	55.5	73.8	16.8	20.2	26.0
16.5	160.9	176.8	189.8	49.2	66.0	87.6	17.4	21.1	26.8	154.0	166.2	178.8	44.0	56.2	74.3	17.0	20.3	25.9
17	162.9	178.0	190.5	51.3	67.8	88.3	17.7	21.4	26.8	154.7	166.7	179.5	45.0	57.1	75.1	17.2	20.5	25.8
17.5	164.8	179.2	191.1	53.2	69.5	88.9	17.9	21.6	26.8	155.2	167.1	180.1	46.1	57.8	75.9	17.5	20.6	25.7
18	166.3	180.2	191.7	55.2	71.2	89.5	18.1	21.9	26.8	155.7	167.6	180.8	46.9	58.3	76.6	17.7	20.7	25.6

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
