# Peer review of "Growth Charts for Height, Weight, and BMI (6–18 y) for the Tuscany Youth Sports Population"

_ijerph, 2019, doi:10.3390/ijerph16244975_

Round 1

Reviewer 1 Report

General comments:
This paper is a well-articulated document, which aims to compare the prevalence of overweight or obesity conditions in young people who engage in organized sports with the general youth population of Italy.
Although the paper appears satisfactorily structured in its logical sequence and the methodological approach was correctly developed, with a clear report of results, the rationale could be more valorized, as the protocol deserved.
In addition, a general reorganization, especially on the writing style (discussion section), will give more appeal to the manuscript. I can sympathize that English is not the first language of the Authors (and also for me, the reviewer); in some locution, there is the opportunity to perform some edits to "embellish" the draft (e.g. line 119, some grammar mistakes).
I would like to thank the Authors for undertaking this work and hope that the points below will help to strengthen the manuscript.

Major concerns:

Line 55: Gender distribution appears unbalanced, are there any potential issues or source for bias in the statistics?

178: Points of strength should be integrated in the lower paragraph.

Minor concerns:

Line 48: Please revise the sentence, "[...] the second purpose of the study is to propose [...]" sounds bad.

Figure 1: I do not know if this appears blur in my pdf version. A clearer picture should be preferred.

127 and following: Please, revise this sentence that appears not completely clear.

Author Response

General comments:
This paper is a well-articulated document, which aims to compare the prevalence of overweight or obesity conditions in young people who engage in organized sports with the general youth population of Italy.
Although the paper appears satisfactorily structured in its logical sequence and the methodological approach was correctly developed, with a clear report of results, the rationale could be more valorized, as the protocol deserved.
In addition, a general reorganization, especially on the writing style (discussion section), will give more appeal to the manuscript. I can sympathize that English is not the first language of the Authors (and also for me, the reviewer); in some locution, there is the opportunity to perform some edits to "embellish" the draft (e.g. line 119, some grammar mistakes).
I would like to thank the Authors for undertaking this work and hope that the points below will help to strengthen the manuscript.

the authors thank the reviewer for the work done to improve the manuscript and at the same time they are proud of the appreciation addressed

Major concerns:

Line 55: Gender distribution appears unbalanced, are there any potential issues or source for bias in the statistics?

Thank you for this observation. 

Being a retrospective study, data was processed from the archive. The fact that there are more males than females is due to the fact of a greater participation of male in young age in organized sport, in particular in the past.
Also in the latest report, the Italian Institute of Statistics (ISTAT, https://www.istat.it/it/files//2019/10/Report_Stili_di_vita_minori.pdf) reports greater male participation. This difference increases especially after adolescence.
This aspect was discussed in the discussion section:
"Generally, at the highest level of participation in sports activities is at a higher level of participation in organized sports activities is at an age between 6 and 10 years with 59.7% of the population Already in the following age group, those 11-14 years, the females begin to abandon the sport, a phenomenon that then consolidates in the age group 15-17 years [18 ]. "

No source of bias in statistical analysis because the development of growth chart was performed with a separate analysis between gender.

178: Points of strength should be integrated in the lower paragraph.

Thank you. The strengths of the study have been moved to the last paragraph of the discussion section.

Minor concerns:

Line 48: Please revise the sentence, "[...] the second purpose of the study is to propose [...]" sounds bad.

This sentence has been revise as:

"Whether this prevalence differs, the linked purpose of the study is to develop for the first time growth charts for height, weight and BMI in an Italian youth sports population"

Figure 1: I do not know if this appears blur in my pdf version. A clearer picture should be preferred.

The authors thank for this feedback. If the editor ask to the authors a new figure, this will be processed and will be upload in system.

127 and following: Please, revise this sentence that appears not completely clear.

Thank you, the sentence has been specified as follows:

"The surveillance system called "Okkio alla salute" detects the weight status of Italian children aged between 8 and 9 years every 2 years. The latest available data refer to 2016 and 26.9% were overweight and obese in the Tuscany Region [19]. In the same age group, the combined prevalence of both sexes in this sample of young athletes is 18%."

Reviewer 2 Report

Thank you very much for allowing me to review the original article "Growth charts for height, weight and BMI (6 - 18 y) for the Tuscany youth sports population." (IJERPH-667831).
The aim of this study is to trace growth charts by height,
weight and body mass index (BMI) to be applied to the youth sports population. And to verify a lower prevalence of overweight or obesity condition in young people who engage in organized sports compared to the general youth population.
A retrospective study was conducted on 14700 young athletes (10469 male and 4231 female) aged between 6 and 18 years from surveillance carried out during the pre-participation screening of sports eligibility (during 21 years: 1998-2019).
Is it a sample with repetition or without repetition? that is, are the children studied different or are they the same children at different ages?
It is necessary to have the approval of an Ethical Committee, it is not co-correct to claim the declaration of Helsinki, since the data collection is until 2019.
Clarify in the clinical evaluation that during the 21 years of the study the same equipment was used.
Please clarify how the objective has been defined since the anthropometric difference between the population of athletes and the general population is not assessed.
Overall rating:
This study provides the evolution of the growth of children and adolescents who play sports for 21 years. This is a long time and the lifestyle is currently very different. On the other hand, it should be clarified whether they are the same children at different times or are different children, since it will affect the results.
On the other hand, the sport allows to develop the muscle mass that weighs more than fat, anthropometry would not be totally adequate since the folds are the ones that would really inform us of the situation of obesity or not of these children. All these aspects should be discussed in the discussion.

Author Response

Thank you very much for allowing me to review the original article "Growth charts for height, weight and BMI (6 - 18 y) for the Tuscany youth sports population." (IJERPH-667831).
The aim of this study is to trace growth charts by height, weight and body mass index (BMI) to be applied to the youth sports population. And to verify a lower prevalence of overweight or obesity condition in young people who engage in organized sports compared to the general youth population.
A retrospective study was conducted on 14700 young athletes (10469 male and 4231 female) aged between 6 and 18 years from surveillance carried out during the pre-participation screening of sports eligibility (during 21 years: 1998-2019).

The authors thank the reviewer for the work done in order to improve our manuscript.

Is it a sample with repetition or without repetition? that is, are the children studied different or are they the same children at different ages?

The authors expected this observation.

All the subjects who repeated their sport eligibility into this sports medicine center several times over the years were rejected from the second time.
So every case observed must be considered as single individual, so there are 14700 different subjects
.

The exclusion criteria sentence has been re write:

"Exclusion criteria in the analyses were: having already carried out the same visit and being already included in the study sample at a lower age; an age outside the range of ± 6 months compared to the average age of its own stratum, and abnormal values for weight and/or height (5 kg below 3rd or 30 kg above the 97th percentile; 5 cm below 3rd or 5 cm above the 97th percentile). "

It is necessary to have the approval of an Ethical Committee, it is not co-correct to claim the declaration of Helsinki, since the data collection is until 2019.

The ethical statement part was not presented as a correction of the Helsinki declaration.
A separate sentence describes the legislative path that the Italian Public Bodies have promoted through resolutions approved by the Governing Bodies of the Tuscany Region.

The 2014-2018 Regional Prevention Plan (available on the website of the Italian Government, http://www.salute.gov.it/portale/temi/documenti/PNP/Toscana_PRP.pdf) provide prevention-oriented actions and the present project relating to Sports Medicine is at page 130 (N. 08 Project: O-range - Sports medicine to support regional surveillance systems).

The possibility was given to acquire data up to 2019 because the effectiveness of monitoring of the Prevention Plans by these Italian Public Bodies is also carried out in the following years as indicated by the site of the Tuscany Region: https://www.ars.toscana.it/images/pubblicazioni/Rapporti/2018/Programma_di_attivit%C3%A0_2019_e_2020-2021.pdf at pag. 19 "17. Monitoring of the Regional Prevention Plan 2014‐2019"

The ethical problem of the retrospective studies is the confidentiality of personal data coming from the archive. The Tuscany Region therefore requested that the data will be processed in an anonymous way. This request was observed during the data processing through hospital procedures.

A sentence has been added:

"All data has been processed anonymously."

Clarify in the clinical evaluation that during the 21 years of the study the same equipment was used.

This sports medicine center has always used first choice equipment as a regional reference center for sports medical services.
The equipment that the hospital makes available for years has been provided by seca gmbh & Co.
It is certain that in 21 years the models have been updated over time, however the quality of the scales and stadiometers it has always been high.

The sentence has been modified:

"Physical examination focused on the measurement of height and weight by trained personnel, using appropriate equipment (seca gmbh &Co) that has been improved over the years providing ever-greater accuracy. "

Please clarify how the objective has been defined since the anthropometric difference between the population of athletes and the general population is not assessed.

The authors are in agreement with the reviewer.

However, studies of prevalence of overweight and obesity in the same territory on the general population have already been performed. Furthermore, there are still active surveillance systems that regularly record the weight status at which to draw data (OKKIO alla Salute, HBSC surveillance system and the Italian Institute of Statistics ISTAT).

The authors therefore did not consider adding any information to the literature that is not already present including also anthropometric parameters of the general population.

Finally, the aim was to create growth chats for the youth sports population.

Overall rating:
This study provides the evolution of the growth of children and adolescents who play sports for 21 years. This is a long time and the lifestyle is currently very different. On the other hand, it should be clarified whether they are the same children at different times or are different children, since it will affect the results.

The authors are in agreement. Two sentence has been added.

In methods section:

"Exclusion criteria in the analyses were: having already carried out the same visit and being already included in the study sample at a lower age"

In discussion section:

"The first is that the subjects were different from each other, all belonged to the same center and the evaluation methodology was standardized."

On the other hand, the sport allows to develop the muscle mass that weighs more than fat, anthropometry would not be totally adequate since the folds are the ones that would really inform us of the situation of obesity or not of these children. All these aspects should be discussed in the discussion.

The authors are in absolute agreement with the reviewer and actually are working on this aspect in another paper under review. 

Body composition analysis generally is perform by different professional figure from that use percentiles and should be considered an additional evaluation. Furthermore, the definition of obesity is still given based on body weight.

In any case a paragraph has already been included in the discussion section about this issue:

"BMI has some inherent limitations in accurately establishing the overweight condition with the concomitant excess fat [23]. Children and adolescents may participate in sports that favor a particular body shape and young athletes engaged in these sports may desire to gain weight and muscle mass. In line with this notion is that some boys and girls in youth sport could have large lean muscle mass, which could elevate their BMI. On the other side, other sports emphasize a slim or lean physique for aesthetic or performance reasons [24]."

The use of specific young athletes growth chart should reduce the error in identify overweight in specific population. In fact, in conclusion is report :

“The development of these new specific growth charts for the youth sports population could reduce the error risk in the young athlete’s weight status identification.”

Round 2

Reviewer 1 Report

-

Reviewer 2 Report

I have reviewed the new version of the manuscript: "Growth charts for height, weight and BMI (6 - 18 y) for the Tuscany youth sports population" (ijerph-667831), as well as the authors' responses.

I have also carefully read the comments of the other reviewer that I found very appropriate.

I have checked the rapid response of the authors and how they have incorporated the suggestions into the manuscript. Those recommendations that have not followed have justified it, as well as clarifying those doubts that survived in the previous manuscript.

All of which leads me to positively assess the work that could be a reference to assess the growth of young athletes.